# Cross-Regional Pollination Behavior of *Trichoplusia ni* between China and the Indo-China Peninsula

**DOI:** 10.3390/plants12213778

**Published:** 2023-11-06

**Authors:** Xianyong Zhou, Huiru Jia, Haowen Zhang, Kongming Wu

**Affiliations:** 1Xianghu Lab, Hangzhou 311258, China; zhouxy160721@163.com; 2Guangdong Laboratory for Lingnan Modern Agriculture, Guangzhou 510640, China; 3State Key Laboratory for Biology of Plant Diseases and Insect Pests, Institute of Plant Protection, Chinese Academy of Agricultural Sciences, Beijing 100193, China; jhuiru@163.com (H.J.); marszhang_0@163.com (H.Z.)

**Keywords:** *Trichoplusia ni*, transoceanic migration, palynology, pollination behavior

## Abstract

Noctuid moths, a group of “non-bee” pollinators, are essential but frequently underappreciated. To elucidate their roles in cross-regional pollination, this study selected the agriculturally significant species, cabbage looper (CL) *Trichoplusia ni*, as a representative model. From 2017 to 2021, this study was conducted on Yongxing Island, situated at the center of the South China Sea. We investigated the flower-visiting activities of CL, including its occurrence, potential host species, and geographic distribution in the surrounding areas of the South China Sea. First, the potential transoceanic migratory behavior and regional distribution of CL were systematically monitored through a comprehensive integration of the data obtained from a searchlight trap. The transoceanic migratory behavior of CL was characterized by intermittent occurrence, with the major migratory periods and the peak outbreak yearly. Furthermore, trajectory analysis confirmed the ability of CL to engage in periodic, round-trip, migratory flights between Southeast Asian countries and China. More importantly, an observation of pollen on the body surface demonstrated that 95.59% (130/136) of the migrating individuals carried pollen. The proboscis and compound eyes were identified as the primary pollen-carrying parts, with no observable gender-based differences in pollen-carrying rates. Further, identifying the pollen carried by CL using morphological and molecular methods revealed a diverse range of pollen types from at least 17 plant families and 31 species. Notably, CL predominantly visited eudicot and herbaceous plants. In conclusion, this pioneering study has not only revealed the long-distance migration activities of these noctuid moths in the East Asian region but also provided direct evidence supporting their role as potential pollinators. These findings offer a critical theoretical basis to guide the development of scientific management strategies.

## 1. Introduction

Plants and insects, through their multifaceted interactions, represent the most crucial components of terrestrial ecosystems. One of the most critical interactions is plants’ provision of food and habitat to insects, while insects act as the primary pollinators to facilitate the pollination of plants [1]. This symbiotic relationship contributes to maintaining the ecological balance, improving environmental quality, and preserving biodiversity [2,3]. Therefore, a fundamental prerequisite for elucidating this relationship is to clearly define the host-plant range of specific insect pollinators [4]. However, research on insect-based pollination has predominantly focused on bees [5], while studies on pollination by “non-bee” insects are relatively scarce despite mounting evidence indicating that their importance in pollination is no less [6,7].

Notably, noctuid moths are a recognized category of nocturnal, primary insect pollinators, but they are often overlooked due to their plant-damaging characteristics [8]. The family Noctuidae represents the largest group in the order Lepidoptera and is characterized by the nocturnal activity of the adults. Additionally, noctuid moths exhibit varying nutritional requirements across different developmental stages, leading to differences in host-plant ranges between the larvae and adults [9]. The larvae cause damage by feeding on the host plants, while the adults visit flowers to supplement their nutritional needs. Previous research on noctuid moths has primarily focused on their occurrence patterns and pest control measures, with limited attention given to their pollination-related activities [10]. Therefore, studying the host-plant range of moths of the family Noctuidae can provide a scientific basis for the cultivation of host plants; the rational layout of crop areas, increasing crop yields; the maintenance of refuges; and highlighting their critical function of pollination.

Identifying pollen is an effective method for determining the host-plant range [11,12,13]. When insects visit flowers, pollen adheres to their body surfaces, and when the carried pollen falls onto the stigma of the same plant species, insects fulfill their role as a pollination medium for the plants. Hence, the host-plant range during the adult stage of insects can be inferred by identifying the pollen attached to the body surface. According to the study of Liu et al. [14], an examination of the pollen on the surface of *Agrotis ipsilon* revealed correlations between the carried pollen and various factors, including gender, site of adhesion, migration period, and the species of the source plants. Similarly, Xu et al. [15] revealed that cotton bollworm populations, trapped on various collection dates and at different sites, carried different pollen types, which were checked using scanning electron microscopy. Furthermore, insects in the family Noctuidae undergo a complete metamorphosis to develop into adults, and their bodies may retain the remnants of host-plant material fed on by the larvae. Therefore, molecular techniques such as DNA barcoding can be used to assess the range of host plants consumed by these insects of the family Noctuidae during their larval and adult stages [16]. Additionally, in a study conducted by Jia et al. [13], the analysis of the composition of the remnants of host-plant material inside the body of 20 geographical populations of *Episyrphus balteatus* in China revealed 1012 host plant species.

This study used cabbage looper (CL) *Trichoplusia ni* as a representative case to better illustrate the importance of the insects of the family Noctuidae in cross-regional pollination. CL is a species of noctuid moth characterized by a specialized migratory behavior and the ability to feed on various crops, especially those belonging to the family Brassicaceae, making it a significant agricultural pest [17]. Under favorable atmospheric conditions, CL is capable of wind-assisted flight covering distances of 500–700 km each night [18]. In North America, during the spring and summer seasons, CL migrate northward along the western coast, with individuals traveling as far north as Canada from the southern regions, including Mexico and Arizona in the United States. During the autumn season, subsequent generations of CL populations migrate back to the warmer southern areas for overwintering. In addition, Lingren et al. [19] detected pollen on the body surface of CL and revealed that all CL individuals carried pollen. Among them, 80% of individuals had visited two or more host plants, with the highest number of pollen types carried by any individual being six. Consequently, the migratory flights of CL have the potential to facilitate cross-regional pollination in multiple crops.

However, research on CL in the Asian region is relatively scarce and thus requires further investigation. Therefore, hypotheses were proposed that sought to explore (1) the potential for the long-distance flight of CL in the Asian region, (2) the potential of CL to facilitate cross-regional pollination in host plants, and (3) the relationship between the pollen-carrying rate and factors such as year, season, and gender in the event that CL indeed contributed to cross-regional pollination for host plants. The carried pollen was analyzed to identify the host plants involved, the characteristics of these host plants, and specific host plants with distinctive features that could indicate the source regions of these insects. To achieve these objectives, this study initially employed a searchlight on Yongxing Island to observe and record the transoceanic migratory behaviors of the insects. Second, trajectory simulations were utilized to model their migration paths. Finally, a combination of scanning electron microscopy and DNA barcoding was used to analyze the pollen found on the surface of CL individuals, thereby identifying the host plants visited by CL.

## 2. Results

### 2.1. Seasonal Population Dynamics of Migratory CL

Between 2017 and 2021, CL individuals were consistently captured using a searchlight trap, with 173 individuals captured over the five-year period (Figure 1). However, field surveys indicated the absence of CL larvae at the monitoring sites, suggesting that the CL individuals collected within the searchlight trap were exclusively a migratory population engaged in transoceanic flights. The highest capture of 125 individuals occurred in 2018. In 2019, it was a year of moderate capture with 31 individuals. Moreover, the years of 2017, 2020, and 2021 were characterized by low numbers of captures, with biomass counts of 11, 4, and 2 individuals, respectively. Statistical analysis of the results indicated that the biomass of CL was highest in 2018, with a significant difference observed (F_4, 1554_ = 3.775; *p* = 0.005). Although no significant differences were observed in biomass in the remaining four years, there was a significant interaction between year and month (F_40, 1554_ = 3.405; *p* < 0.001).

The daily population dynamics results suggested that significant fluctuations, characterized by sudden increases or decreases, were observed from mid-September to early October, which corresponded to the primary migration period. Occasional occurrences or non-occurrences were observed during the remaining period (Figure 1). The peak periods in 2017–2019 were observed on 19 September (three individuals), 5 October (36 individuals), and 20 September (eight individuals), with none observed in 2020 and 2021.

### 2.2. Seasonal Pattern of Sex Ratio of Migratory CL

Overall, the biomass of male CL individuals (*n* = 125) was higher than that of female CL individuals (*n* = 47). Notably, all the individuals observed in January were female. The proportion of females was 54.90% ± 10.76 [± SEM] in September and was the same for the subsequent months. In November, the biomass of females (50% ± 22.36) was equal to that of males. However, in the remaining months, the biomass of females was lower than that of males (February: 33.33% ± 33.33; July: 0.00% ± 0.00; October: 23.73% ± 7.87; December: 16.67% ± 16.67). Statistical analyses demonstrated no significant differences in the proportion of female CL across years and seasons (Year: F_4, 34_ = 1.705, *p* = 0.172; Month: F_7, 106_ = 0.969, *p* = 0.437), as well as no significant interactions between year and month (F_6, 34_ = 1.887, *p* = 0.112).

### 2.3. Migration Pattern of Migratory CL

The results of the forward and backward trajectory analyses indicated that CL were capable of periodic, round-trip, migratory flights between Southeast Asian countries such as Cambodia, Laos, Thailand, Vietnam, and the Philippines; and the southern regions of China, including Guangdong, Fujian, and Taiwan provinces. The specific findings are detailed as follows (Figure 2).

In July (summer migration period), the source CL population on Yongxing Island originated from Vietnam (southeastern and south-central regions) and Cambodia (central and eastern regions). The forward trajectory analysis indicated that their migration destination was located in the north-central part of Vietnam.

During the period from September to November (autumn migration period), the trapped population predominantly originated from the coastal regions of Fujian and Guangdong provinces in China and the northernmost part of the Ilocos Region of the Philippines. They also migrated from the coastal regions of Guangxi, Guangdong, Hainan, and Taiwan provinces of China. The primary migration destinations during this period were located in the south-central and north-central regions of Vietnam. Following these regions, the migration extended to the southeastern parts and the Mekong Delta in Vietnam, northeastern Thailand, and the central and southern regions of Laos. Additionally, a small proportion of the migrating population reached parts of the Philippines, specifically Mindoro Island and the Calabarzon region.

During the period from December to February (winter migration period), the migrating population primarily originated in the coastal regions of China, including Fujian and Guangdong provinces, and the southern parts of Taiwan province, as well as the northern parts of Luzon Island in the Philippines. These populations primarily migrated to the south-central and north-central regions of Vietnam. Subsequently, their migration extended to other areas, including the southeastern parts of Vietnam, the Mekong Delta, southern Laos, and the southwestern region of Cambodia, with some impacts on northeastern Thailand.

### 2.4. Results for Surface Pollen Carried by Migratory CL

#### 2.4.1. Pollen-Carrying Body Parts in Migratory CL

The analysis of the pollen on the surfaces of 136 CL individuals captured between 2017 and 2020 revealed that a substantial proportion, 130 out of 136 (95.59%), were found to carry pollen. Pollen primarily adhered to the proboscis (2017: 72.73%, 8/11; 2018: 86.81%, 79/91; 2019: 80.65%, 25/31; 2020: 66.67%, 2/3) and the compound eyes (2017: 36.36%, 4/11; 2018: 25.27%, 23/91; 2019: 41.94%, 13/31; 2020: 33.33%, 1/3). A minimal amount of pollen was found on the antennae (2018: 1.10%, 1/91), while no pollen was detected on the labial palps, thorax, fore/mid/hind legs, or abdomen (Figure 3A). Statistical analysis indicated a significant correlation between the pollen-carrying rate and the attachment parts on the body (Table 1).

#### 2.4.2. Seasonal Pollen-Carrying Rate of Migratory CL

The pollen-carrying rates for CL from 2017 to 2020 were 81.82% (9/11), 97.80% (89/91), 93.55% (29/31), and 100.00% (3/3), respectively. In July (summer), September to November (autumn), and December to February (winter), the pollen-carrying rates were 85.57% ± 3.71, 82.18% ± 1.47, and 100.00% ± 0.00, respectively (Figure 3B). Over the four years, the lowest pollen-carrying rate was recorded in October 2017 (75.00%), while in other months, the pollen-carrying rate remained ≥ 80.00% (Figure 3C).

#### 2.4.3. Sex Differences in Pollen-Carrying Rates

In 2017, a total of 11 CL individuals (six females and five males) were examined, with pollen-carrying rates of 45.45% (5/11) for females and 36.36% (4/11) for males. No significant difference was identified between the two genders (χ^2^ = 0.111, *df* = 1, *p* = 1.000). In 2018, a total of 91 CL individuals (23 females and 68 males) were examined. The pollen-carrying rate in females (24.18%, 22/91) was significantly lower than that of males (73.63%, 67/91) (χ^2^ = 22.753, *df* = 1, *p* < 0.001). In 2019, a total of 31 CL individuals (eight females, 22 males, and one unknown) were subjected to the analysis. The pollen-carrying rate of females (19.35%, 6/31) was significantly lower than that of males (70.97%, 22/31) (χ^2^ = 9.143, *df* = 1, *p* = 0.004). In 2020, three CL individuals (two females and one male) were examined, with a pollen-carrying rate of 66.67% (2/3) in females and 33.33% (1/3) in males. There were no significant differences between the two genders (χ^2^ = 0.333, *df* = 1, *p* = 1.000). Over the four years, no significant difference in pollen-carrying rates was observed between the males and females (t = −1.359, *df* = 3, *p* = 0.267). The mean pollen-carrying rates for all individuals, females, and males, were 92.49% ± 4.12, 38.91% ± 10.85, and 53.57% ± 10.84, respectively (Figure 3D). Similarly, there were no significant gender-specific differences in the pollen-carrying rates of the proboscis and compound eyes in CL (Figure 3E,F). Additionally, the differences in the pollen-carrying rates were not significant between the proboscis and compound eyes (Figure 3G–I).

#### 2.4.4. Host Plants Inferred from Pollens Carried by Migratory CL

In total, 31 types of pollen were identified through morphological observations and molecular barcoding (Figure 4). Among them, 15 pollen types were characterized at the species level, seven at the genus level, one at the family level, and ten as unknown plant pollen. During the autumn season, the CL population carried 22 recognized pollen types with the sources being *Helianthus annuus*, *Aster tataricus*, *Tripolium vulgare*, *Chrysanthemum zawadskii*, *Rubia cordifolia*, *Humulus scandens*, *Bidens alba*, *Rinorea bengalensis*, *Bougainvillea spectabilis*, *Impatiens balsamina*/*I. walleriana*, *Brassica carinata*/*B. juncea*/*B. nigra*, *Melia azedarach*, *Amorpha fruticosa*, *Platanus acerifolia*, *Scaevola taccada*; the genera *Ligustrum*, *Heliotropium*, *Alisma*, *Festuca*, *Eucalyptus*, and *Asparagus*; and the family Violaceae Batsch, making it the most diverse in terms of pollen types. In the winter, they carried five known pollen types, with the sources being *Aster tataricus*, *Rubia cordifolia*, *Bidens alba*, *Helianthus annuus*, and the genus *Phaseolus*, as detailed in Table 2.

#### 2.4.5. Characteristics of Host Plants Producing Pollen Carried by Migratory CL

The analysis of pollen source-plant species visited by migrating CL adult insects from 2017 to 2020 revealed that CL exclusively visited plants belonging to Angiospermae. They preferred eudicot plants over monocot plants, with a highly significant difference observed (χ^2^ = 15.565, *df* = 1, *p* = 0.000), and showed a preference for herbaceous plants over woody plants (χ^2^ = 1.087, *df* = 1, *p* = 0.405) (Figure 5). Among the 23 host plants identified, the majority were widely distributed in China and Southeast Asia, with only a few found exclusively in China. Pollen from plant species unique to Southeast Asia was not observed.

## 3. Discussion

Yongxing Island is situated in the South China Sea, far from agricultural areas, without arable land or agrarian activities. The environmental conditions on the island do not support the reproduction of local agricultural pests. Daily field surveys have also indicated the absence of the larvae of agricultural pests on the island. Therefore, the population of CL adults attracted using a searchlight trap can be considered migratory populations originating from other areas, providing direct evidence for the migration of CL. Furthermore, Zhou et al. used a searchlight trap to identify the structure of the nocturnal aerial insect community on Yongxing Island and employed a combination of techniques and methods, including searchlight trap, molecular identification, ovarian dissection, and trajectory analysis, to elucidate the cross-South China Sea migration patterns of invasive pests such as *Spodoptera frugiperda* [20]. Thus, it can be concluded that Yongxing Island is an ideal field site for monitoring the migration of wild insects.

The transoceanic migration of CL primarily occurs in September and October, especially in October, with sporadic occurrences during the remaining months. This pattern may be attributed to several factors. First, from March to May (spring), rice is mainly cultivated in the Indo-China Peninsula, which is not a suitable host for CL. Second, from June to August (summer), the temperatures in the year-round breeding areas exceed the optimal temperature range for CL, resulting in smaller population sizes in those regions [18]. Third, similar to *Autographa gamma*, CL is more adapted to high-latitude temperate regions, and the year-round breeding areas represent a population bottleneck. Consequently, the size of the returning population during autumn is higher than that of the migrating population in summer [21,22]. Fourth, CL exhibits an intermittent occurrence pattern. Fifth, the monitoring site is located in the central South China Sea, far from agricultural areas of the mainland, creating a natural “death trap” for CL. Therefore, the probability of capturing CL individuals is enhanced during transoceanic migration when a sufficient number of individuals are involved. Considering the factors mentioned above, the migration of CL primarily occurs from September to October. In contrast, under similar temperatures, the overwintering range of CL in the United States is more extensive. It includes vast areas for planting vegetables, providing ample food sources for the overwintering population. This phenomenon results in significant damage to the horticulture industry in the United States [18]. However, during the same period in China, the main crops cultivated are rice and wheat, which are unfavorable hosts for CL, thereby effectively avoiding significant damage. This is also one of the key reasons why CL has been relatively overlooked in China.

Trajectory analysis is one of the most potent tools used currently to trace the source and migration patterns of migratory insects. It reveals the potential source areas, migration routes, and destinations of these insects [23]. This approach assists institutions and farmers in proactively understanding the migratory dynamics of insect pests and implementing effective pest management strategies. This approach has been validated in agricultural pests such as *S. frugiperda* and *Helicoverpa* spp. [24,25,26]. In this study, trajectory simulations revealed the periodic, round-trip migratory pattern of CL between the Indo-China Peninsula (including Thailand, Vietnam, Laos, and Cambodia), South China, and the Philippines. In July, CL from the southern regions of the Indo-China Peninsula (Vietnam and Cambodia) migrated northward, following the southerly winds. After September, with a seasonal change in the wind patterns, the atmospheric circulation over Asia is dominated by continental high-pressure systems. The offspring of CL populations in China and the Philippines then take advantage of the northerly winds to migrate to the countries of the Indo-China Peninsula for overwintering. This seasonal round-trip migratory pattern is similar to the cross-migration over the South China Sea observed in other pests such as *S. frugiperda* [20]. This comparative analysis underscores the similarity in migration patterns among various insect species in the same migration area [27]. It further confirms the potential entry of pests from Southeast Asian countries into China through the South China Sea, providing valuable additional information for investigating the initial pathways of foreign insect sources migrating into China. Additionally, it offers data support for international cooperation in “green” pest control.

Despite posing a threat to the ecosystems of different regions, insect migration enables cross-regional pollination in host plants [28]. As an excellent natural marker, pollen is characterized by its small size, stable properties, and storage resistance [29]. Moreover, the type of pollen carried is directly associated with the food habits of the insects, thereby avoiding the detrimental effects of artificial marking. Pollen does not impact the biological parameters of insects, such as lifespan, flight, and reproduction. Through the identification of pollen on the body surface, we found that almost all migrating CL individuals carried pollen, and the pollen was primarily attached to the proboscis and compound eyes. These findings suggest that CL proactively visit flowers but not incidentally, thereby supplementing the nutritional needs of CL for completing a long-distance migration across the sea. Furthermore, several studies have demonstrated that pollen is mainly found on the proboscis and compound eyes [30,31,32,33,34]. Moreover, the high carrying rate also implies that CL is likely to be a pollinator with considerable potential. Similarly, Xu et al. [15] analyzed the pollen carried by *H. armigera* and demonstrated a high pollen-carrying rate in the populations involved in transoceanic migration. Lingren et al. [19] examined the pollen on the body surface of CL in the United States and revealed that all individuals carried pollen, with 80% of them having visited two or more host plants. Furthermore, multiple studies have demonstrated that pollen is mainly found on the proboscis and compound eyes [30,35]. Based on these observations, it is advisable to conduct research specifically on the pollen from the proboscis and compound eyes of insects to simplify the detection process and ensure the reliability of results.

Through morphological observations and molecular barcoding of pollen, 31 species of host plants belonging to 17 families were identified, with a preference of CL for visiting eudicot and herbaceous plants. The food type and feeding sites of most phytophagous Lepidoptera differ between the adult and larval stages. The larval food plants are mainly determined by the female insects during oviposition [9], and they primarily consume tender parts such as leaves. Before flight, adult insects need to rapidly complete the process of nutrient supplementation, and the sugars such as glucose and fructose found in floral nectar can effectively fulfill this requirement [15]. Utilizing this characteristic of CL in agricultural productivity allows for the rational planning of crop layouts, the reduction of damage, and the simultaneous maximization of the pollination function of CL.

Most of the 23 host plant species recognized are widely distributed in China and Southeast Asia, with only a few being exclusive to China. No observations of pollen from plants unique to Southeast Asia have been recorded. Notably, only one CL individual was attracted during the summer season (on 25 July), with the rest collected in the autumn and winter seasons. Unfortunately, due to a failure in collection, photography and identification of the specimen were impossible. Based on the information mentioned above, it can be suggested that the host plants endemic to China serve as the source regions of CL from September to February of the following year (autumn/winter), providing direct evidence for CL migration.

Due to the intolerance of CL to high temperatures and the limited sample size, especially from March to August (spring/summer), the research on the northward migration of CL in this region conducted in this study is not comprehensive. The migratory dynamics of CL on Yongxing Island need to be further monitored, and the Southeast Asian host plants that it visits should also be investigated in the future. The plant-related information, which includes the geographical indicative characteristics, may be used as potential sources for CL breeding using backward trajectory analysis. Furthermore, various techniques and methods, such as indoor tethered flights, ovary dissection, stable isotopes, genetic differentiation analysis, and insect radar, should be employed to further study CL migration. In the past, due to insufficient knowledge, noctuid moths were often unilaterally classified as pests and controlled through methods such as light traps, attractants, insecticides, natural enemies, and genetically modified crops [36,37,38]. However, this study highlighted that CL is a prominent flower visitor that possibly facilitated gene exchange among the plants in the region around the South China Sea. Based on the latest progress in research into flower-visiting ability by noctuid moths and the results of our team’s study on the migratory behaviors of noctuid moths in cross-regional pollination, it can be emphasized that these insects may have highly promising pollination capabilities and play a crucial role in maintaining the functions of ecosystems [39]. Therefore, it is necessary to strengthen the research effort to better balance the utilization of the possible pollination activity of noctuid moths with comprehensive pest control measures.

In summary, this study integrated multiple technical approaches and methods, including searchlight trap, trajectory simulation, pollen marking, and DNA barcoding, to systematically monitor the trans-South China Sea flights of CL for seasonal migration. This study aimed to clarify the characteristics of the seasonal occurrences in the year-round breeding area and the north–south round-trip migration patterns of CL. It was found that adult CL were capable of visiting 31 plant species belonging to 17 families. These findings provide an essential scientific basis for establishing regional monitoring, early warning, and control systems for migratory insect pests in China and the countries surrounding the South China Sea.

## 4. Materials and Methods

### 4.1. Study Site

The monitoring site for this study was Yongxing Island, with an altitude of 5 m above mean sea level, located at 16°49′59″ N, 112°19′55″ E. Yongxing Island is situated in the eastern part of the Xisha Islands in the South China Sea. It is far from the inland agricultural areas (~280 km south of Hainan Island, China; ~410 km east of Vietnam; and ~780 km west of the Philippines). The island lacks arable land, which does not allow agricultural activities or support the proliferation of local agricultural pests. This environment effectively eliminates the interference by non-migratory insect populations, making it an ideal natural location for studying insect migration.

### 4.2. Monitoring of Searchlight System

A searchlight was deployed at the monitoring station on Yongxing Island, as described in the above Section “Study sites.” The design was based on the specifications reported by Zhou et al. [20]. The searchlight consisted of a JLZ1000BT 1 KW metal halide lamp (Shanghai Yaming Lighting Co. Ltd., Shanghai, China). The vertical light beam generated by this lamp rendered a significant attractive effect on the airborne insect populations within an altitude of 500 m. During the monitoring period, from 28 February 2017 to 27 September 2021, the searchlight was operated daily from sunset to sunrise, except in cases of equipment malfunction, power outages, or heavy rain. The collection net at the bottom, made of 60-mesh nylon mesh with length × width × height dimensions of 30 × 30 × 40 cm, was replaced at a fixed time daily. After collection, the insects were stored in a –20 °C freezer for 4 h. The insects were then classified manually based on their morphological characteristics. The corresponding species-specific trapping data were recorded for the analysis of population dynamics. Throughout the entire study period, daily field observations were conducted to check the occurrence of CL larvae on potential wild host plants to further confirm the presence or absence of a local population at the monitoring site.

### 4.3. Trajectory Simulation

The Final Analysis (FNL) data from the National Centers for Environmental Prediction (NCEP) in the United States were used as the initial field data and to set up the boundary parameters for initiating the design of the Weather Research and Forecasting (WRF) model. These data were collected at a time interval of 6 h and a spatial resolution of 1.0° × 1.0°. Meteorological elements at a resolution of 10 × 10 km were the output collected every hour (Table 3).

The biological parameters used for the trajectory analysis were configured as follows: (1) CL takeoffs predominantly occur at sunset [40,41]. Consequently, the simulation time ranged from 06:00 h of the next day to 18:00 h on the same day, with trajectories simulated each hour, resulting in a total of 72 trajectories (12 h × 6 altitudes). (2) CL was assumed to adopt a downwind flight posture [42,43]. (3) The altitudes of the simulated flights for CL were 250, 500, 750, 1000, 1250, and 1500 m [44,45]. (4) Flight simulations were terminated when the temperature dropped below 10 °C [20]. (5) The duration of the simulated migratory flights did not exceed 36 h [46].

### 4.4. Detection of Pollen on the Body Surface

#### 4.4.1. Microscopic Examination and Scanning Electron Microscopic Imaging

Pollen collected from the various body parts of the CL individuals, such as proboscis, compound eyes, antennae, labial palps, thorax, fore/mid/hind legs, and abdomen, carried from 28 February 2017 to 9 November 2020, was examined and selectively gathered under a SZX16 stereomicroscope (Olympus, Pittsburgh, PA, USA). The collected pollen was attached to a conductive platform covered with double-sided adhesive tape, and the date and the pollen-carrying sites of the samples were recorded. The images of the pollen were captured using a Regulus 8100 scanning electron microscope (Hitachi, Tokyo, Japan) after gold spraying.

#### 4.4.2. Molecular Analysis

DNA barcoding of individual pollen grains was conducted to enhance the accuracy of species identification, which was previously photographed. The genomic DNA of single pollen grains was extracted following the method outlined by Chen et al. [47]. In brief, a single pollen grain was placed in a 0.2 mL PCR tube containing 5 µL of lysis solution (0.1 M NaOH + 2% Tween-20) and incubated at 95 °C for 17.5 min in a GeneAmp PCR System 9700 (Applied Biosystems, Foster City, CA, USA). Subsequently, 5 µL of Tris-EDTA (TE) buffer was added as a template for subsequent PCR amplification. The pollen was identified using the mitochondrial intergenic spacer region, ITS2 [48]. The target segment was amplified using the Platinum SuperFi II Master Mix (2X) (Invitrogen, Thermo Fisher Scientific, Waltham, MA, USA) and the primers ITS-p3 and ITS4. The reaction volume was 20 µL, and the amplification conditions were set as follows: initial denaturation at 98 ℃ for 3 min, followed by 35 cycles of denaturation at 98 ℃ for 10 s, annealing at 60 ℃ for 10 s, and extension at 72 ℃ for 10 s; and final extension at 72 ℃ for 10 min. The PCR products were purified using a gel extraction kit EG101-02 (TransGen, Beijing, China), and the recovered products were ligated to the pEasy-T3 vector (TransGen, Beijing, China). Randomly selected positive clones were sequenced using Sanger sequencing (Zhongmei Taihe Bio-Technology Co. Ltd., Beijing, China).

#### 4.4.3. Pollen and Host Plant Identification

The pollen grains were identified through their morphological characteristics, molecular identification results, and geographical distribution patterns. The gene sequences were compared to the gene sequences available in the National Center for Biotechnology Information (NCBI; https://www.ncbi.nlm.nih.gov/search/accessed on 8 October 2023) database using the BLASTn online search program. If the sequence matched with a single species, or multiple species within the same genus, or multiple genera within the same family, it was considered to correspond with the respective species, genus, or family. Unmatched sequences were labeled as “unidentifiable” [49]. Based on the molecular identification data, further comparisons were made between the morphology of the pollen and the scanning electron microscopic images of the pollen grains from the Chinese flora [50,51] or the online palynological database (https://www.paldat.org/ accessed on 5 October 2023). Finally, the distribution of the identified plants was determined using the Flora of China species library and the Plant Science Data Center (https://www.plantplus.cn/cn accessed on 5 October 2023).

### 4.5. Statistical Analysis

The normality and homogeneity of the variances in CL biomass and the proportion of female CL were assessed using the Shapiro–Wilk and Levene’s Tests, respectively. After passing the tests, differences in CL biomass and the proportion of female CL over years and months were ascertained using a general linear model, with months as fixed and years as random factors [52]. Percentage data were analyzed after square-root-arcsine transformation. In cases of significant differences, multiple comparisons were performed using Tukey’s Honestly Significant Difference (HSD) test. Paired *t*-tests were employed to analyze the interannual differences in the pollen-carrying rates. The chi-square test, a non-parametric test, was used to compare the sex ratio of CL in each month and the theoretical sex ratio (1:1), and the differences in the pollen-carrying rates across various body parts and gender, as well as in the same gender but different body parts (proboscis and compound eyes). If the total sample size (n) was ≥40 and the expected values (T) were ≥5, the chi-square test was conducted using the Monte Carlo simulation method. If n was <40 or T was <1, the chi-square test was performed using the exact probability method. All statistical analyses were carried out using SPSS Statistics Version 25.0 (IBM, Armonk, NY, USA).

The trajectory endpoints were clustered using the kernel density estimation method, with an output pixel size of 0.01. The default best search radius was used, and the results of clustering were divided into five categories using the natural breaks (Jenks) method, where points closer to the kernel corresponded to higher weights [53,54,55].

## Figures and Tables

**Figure 1 plants-12-03778-f001:**
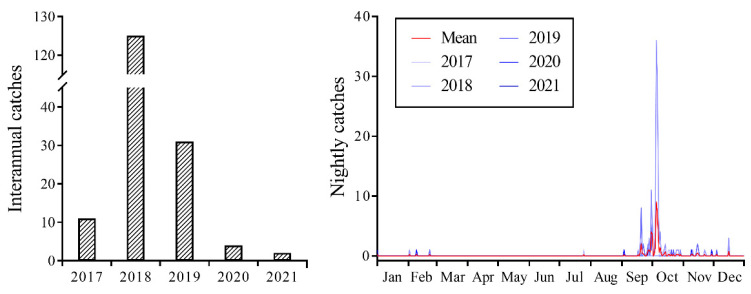
Seasonal population dynamics of *Trichoplusia ni* trapped on Yongxing Island using searchlight from 2017 to 2021.

**Figure 2 plants-12-03778-f002:**
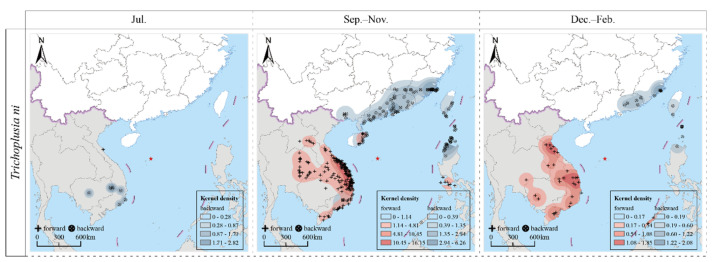
Trajectories of migrating population of *Trichoplusia ni* during 2017–2021.

**Figure 3 plants-12-03778-f003:**
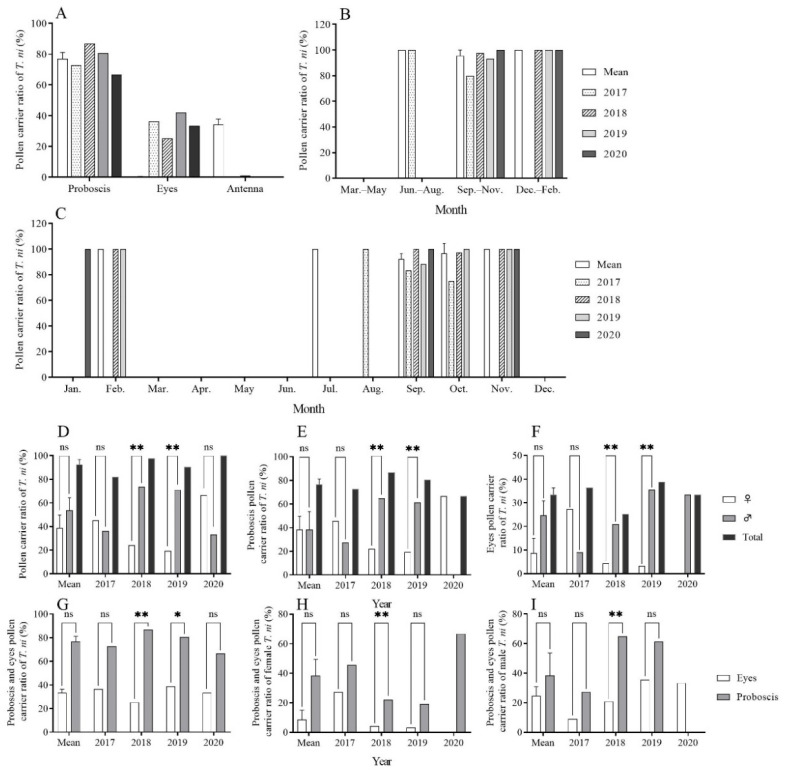
Pollen-carrying rates for trapped *Trichoplusia ni* using searchlight on Yongxing Island from 2017 to 2020. Note: *, ** and ns indicate *p* < 0.05, *p* < 0.01 and no difference, respectively.

**Figure 4 plants-12-03778-f004:**
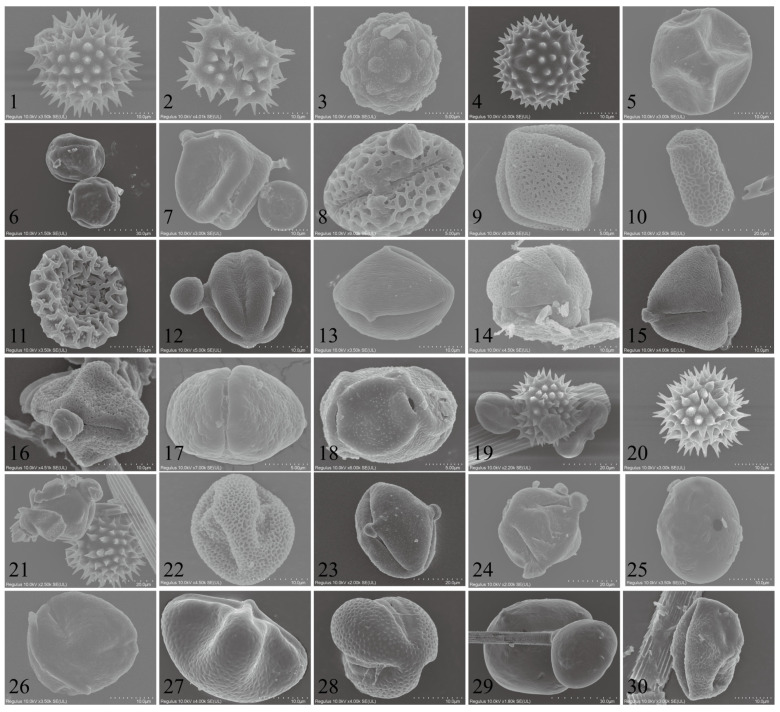
SEM microphotographs of the examined pollen grains adhered to trapped *Trichoplusia ni* individuals.

**Figure 5 plants-12-03778-f005:**
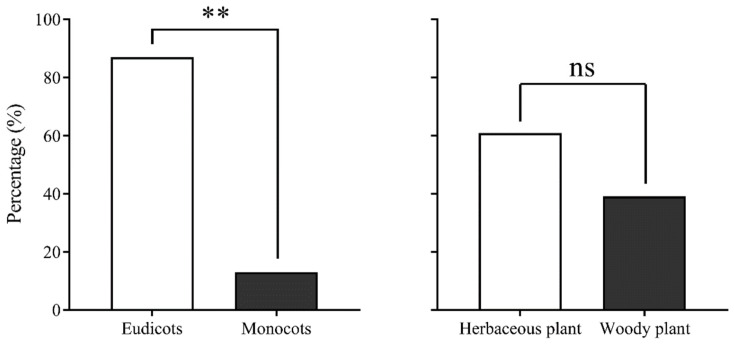
Characteristics of pollen source plants of migrating *Trichoplusia ni* on Yongxing Island. Note: ** and ns indicate *p* < 0.01 and no difference, respectively.

**Table 1 plants-12-03778-t001:** Differences in pollen-carrying rate in *Trichoplusia ni* between pollen-carrying body parts.

Year	Pollen-Carrying Body Parts	χ^2^	*df*	*p*
2017	Proboscis vs. Compound eyes	1.333	1	0.388
2018	Proboscis vs. Compound eyes	30.745	1	<0.001
	Proboscis vs. Antennae	76.050	1	<0.001
	Compound eyes vs. Antennae	20.167	1	<0.001
2019	Proboscis vs. Compound eyes	3.789	1	0.073
2020	Proboscis vs. Compound eyes	0.333	1	1.000

**Table 2 plants-12-03778-t002:** Types of pollen carried on trapped *Trichoplusia ni* individuals.

No.	Identification	Trapped Periods	Class	Stem	Flowering Periods
1	*Aster tataricus*	February, September–November	Eudicots	Herb	July–September
2	*Tripolium pannonicum*	September	Eudicots	Herb	June–December
3	*Alisma* L.	October	Monocots	Herb	May–October
4	*Chrysanthemum zawadskii*	October	Eudicots	Herb	June–November
5	*Festuca* L.	September	Monocots	Herb	June–September
6	*Melia azedarach*	October	Eudicots	Xylophyta	April, May
7	*Heliotropium* L.	September–October	Eudicots	Herb	February–October
8	*Ligustrum* L.	October	Eudicots	Xylophyta	April–August
9	*Brassica carinata*/*B. juncea*/*B. nigra*	October	Eudicots	Herb	April, May
10	*Impatiens balsamina*/*I. walleriana*	September	Eudicots	Herb	July–October
11	*Bougainvillea spectabilis*	September–October	Eudicots	Xylophyta	June–December
12	*Rubia cordifolia*	October	Eudicots	Herb	July–September
13	Violaceae	September	Eudicots	Xylophyta	-
14	*Amorpha fruticosa*	October	Eudicots	Xylophyta	May, June
15	*Rinorea bengalensis*	October	Eudicots	Xylophyta	March–August
16	*Asparagus* L.	October	Monocots	Herb	May–August
17	*Eucalyptus* L’Hér.	October	Eudicots	Xylophyta	March, April
18	*Humulus japonicus*	October	Eudicots	Herb	July–September
19	*Helianthus annuus*, *Rubia cordifolia* and *Phaseolus* L.	February	Eudicots	Herb	June–September
20	*Bidens alba*	February, September	Eudicots	Herb	August, September
21	*Helianthus annuus* and Unknown	September–October	Eudicots	Herb	July–September
22	*Platanus × acerifolia*	October	Eudicots	Xylophyta	April–May
23	*Scaevola taccada*	September–October	Eudicots	Xylophyta	April–December
24–30	Unknown	September–October			

**Table 3 plants-12-03778-t003:** Scheme and parameters of WRF model.

Item	Domain 1
Location	16°50.1′ N, 112°19.8′ E
Number of grid points	200 × 200
Distance between grid points	10
Layers	32
Map projection	Lambert
Microphysics scheme	Thompson
Longwave radiation scheme	RRTMG
Shortwave radiation scheme	RRTMG
Surface layer scheme	Monin–Obukhov
Land surface scheme	Noah
Planetary boundary layer scheme	Mellor–Yanmada–Janjic
Cumulus parameterization	Tiedtke
Simulated time	Every 36 h

## Data Availability

Not applicable.

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
