# Peer review of "Cross-Regional Pollination Behavior of Trichoplusia ni between China and the Indo-China Peninsula"

_plants, 2023, doi:10.3390/plants12213778_

Round 1
Reviewer 1 Report
Comments and Suggestions for Authors
This is an interesting contribution that, in my opinion, needs some improvements. First, the authors use the term "host plant" in a way that is actually confused with plants that the moths may actually visit/pollinate. Please, along the text, clarify when authors refer to host plants (where larvae develop) or plants that moths visit/pollinate.
Another important point: since there is no direct evidence of pollination (moths should actually contact both, the stamens and the gynoecium of all these plants to be safely considered pollinators), I strongly recommend avoiding stating that these Lepidoptera pollinate these plants. Without clear evidence indicating that moths deposit these pollens onto the respective stigmas, they should be considered flower visitors instead. So, briefing, consider the moths as flower visitors.
The terms "Monocotyledons" vs "dicotyledons" (ex: Table are used throughout the text (Ex: Figure 5, Table 2). This plant classification is outdated (See Judd et al., 2023). Whereas the Monocotyledons are a monophyletic group, the so-called "dicots" aren't. By checking the plants, it is clear that they are either monocots or eudicots (plants with 3-colpate pollen or types derived from this). I strongly recommend changing the USD nomenclature for a more modern one (See Judd et al., 2023 or APWEB, 2023).
All in all the results are interesting and comply with precedent reports indicating that Noctuidae moths mostly carry the pollen onto the eyes or the proboscis; the only non-scaly places of their bodies. Notably, Noctuidae carry the pollinia of several terrestrial orchids onto the eyes (ex: Singer & Cocucci, 1997) or onto the proboscis (Singer, 2002).
Comments on the Quality of English Language
I think the English needs moderate editing.
Author Response
Reviewer#1:
This is an interesting contribution that, in my opinion, needs some improvements.
Response: Thank you for your constructive feedback. We have carefully considered each of your comments and have made appropriate revisions to enhance the clarity and accuracy of our manuscript.
Q1: First, the authors use the term "host plant" in a way that is actually confused with plants that the moths may actually visit/pollinate. Please, along the text, clarify when authors refer to host plants (where larvae develop) or plants that moths visit/pollinate.
Response: We regret any confusion caused by our use of the term "host plant." We have clarified our terminology throughout the manuscript to distinguish between 'larval host plants' and 'visited plants' that are involved in adult moth interactions. This distinction ensures a precise understanding of the ecological relationships described.
Q2: Another important point: since there is no direct evidence of pollination (moths should actually contact both the stamens and the gynoecium of all these plants to be safely considered pollinators), I strongly recommend avoiding stating that these Lepidoptera pollinate these plants. Without clear evidence indicating that moths deposit these pollens onto the respective stigmas, they should be considered flower visitors instead. So, briefing, consider the moths as flower visitors.
Response: Your point is well-taken. In the revised manuscript, we have described the moths as "flower visitors" rather than "pollinators," to more accurately reflect their interactions with plants in the absence of direct evidence for pollination.
Q3: The terms "Monocotyledons" vs "dicotyledons" (ex: Table are used throughout the text (Ex: Figure 5, Table 2). This plant classification is outdated (See Judd et al., 2023). Whereas the Monocotyledons are a monophyletic group, the so-called "dicots" aren't. By checking the plants, it is clear that they are either monocots or eudicots (plants with 3-colpate pollen or types derived from this). I strongly recommend changing the USD nomenclature for a more modern one (See Judd et al., 2023 or APWEB, 2023).
Response: Following your recommendation, we have updated the manuscript to replace "Monocotyledons" and "Dicotyledons" with "Monocots" and "Eudicots," respectively. We have made sure this change is consistent throughout the document and reflects the current understanding of plant phylogeny.
Q4: All in all the results are interesting and comply with precedent reports indicating that Noctuidae moths mostly carry the pollen onto the eyes or the proboscis; the only non-scaly places of their bodies. Notably, Noctuidae carry the pollinia of several terrestrial orchids onto the eyes (ex: Singer & Cocucci, 1997) or onto the proboscis (Singer, 2002).
Response: Thank you for your insightful observations on Noctuidae moths’ pollen-carrying behavior. We have incorporated this information into our discussion section, further elucidating the similarities between our findings and well-documented behaviors in other Noctuidae species. Your references are now duly cited, enriching the context of our results.
Q5: I think the English needs moderate editing.
Response: We have undertaken a comprehensive review of the manuscript for language clarity and correctness.
Reviewer 2 Report
Comments and Suggestions for Authors
Although noctuid moths are pollinators in the ecosystem, but their value as pollinators has tended to be underestimated, and there has been little research on them, the authors have obtained a lot of data on one agriculturally important species of moths over a long period of time and have conducted a careful and in-depth analysis of these data. The results not only revealed the long-distance migratory activity of this species in East Asia, but also provided direct evidence supporting its role as a potential pollinator in the ecosystem. Based on the above, I consider this study to be highly novel and important. Furthermore, although the study species positioned as an agricultural pest, this study suggests a pollinator role and very promising pollination capabilities within the ecosystem of this species. Thus, I believe that this study is well worth publication in that it suggests that noctuid moths may play an important role in maintaining the functioning of ecosystems.
Author Response
Reviewer#2:
Although noctuid moths are pollinators in the ecosystem, but their value as pollinators has tended to be underestimated, and there has been little research on them, the authors have obtained a lot of data on one agriculturally important species of moths over a long period of time and have conducted a careful and in-depth analysis of these data. The results not only revealed the long-distance migratory activity of this species in East Asia, but also provided direct evidence supporting its role as a potential pollinator in the ecosystem. Based on the above, I consider this study to be highly novel and important. Furthermore, although the study species positioned as an agricultural pest, this study suggests a pollinator role and very promising pollination capabilities within the ecosystem of this species. Thus, I believe that this study is well worth publication in that it suggests that noctuid moths may play an important role in maintaining the functioning of ecosystems.
Response: We are sincerely grateful for your encouraging evaluation of our research. Your recognition of the novelty and importance of our findings inspires us to continue our work in this underexplored area.
Reviewer 3 Report
Comments and Suggestions for Authors
This paper presents a highly original and captivating study on the cross-regional pollination behavior of Noctuidae moths between China and the Indo-China Peninsula. I have a few minor suggestions for improvement. I recommend implementing references at L57, including other examples where identifying pollen on the body surface of insects has been demonstrated as an effective method for determining the host plant range. These examples can encompass a range of approaches, from molecular to morphological technique. Some potential examples here:
· Galimberti, Andrea, Fabrizio De Mattia, Ilaria Bruni, Daniela Scaccabarozzi, Anna Sandionigi, Michela Barbuto, Maurizio Casiraghi, and Massimo Labra. "A DNA barcoding approach to characterize pollen collected by honeybees." PLoS One 9, no. 10 (2014): e109363.
· Scaccabarozzi, Daniela, Kingsley W. Dixon, Sean Tomlinson, Lynne Milne, Björn Bohman, Ryan D. Phillips, and Salvatore Cozzolino. "Pronounced differences in visitation by potential pollinators to co-occurring species of Fabaceae in the Southwest Australian biodiversity hotspot." Botanical Journal of the Linnean Society 194, no. 3 (2020): 308-325.
Could you also provide further clarification for the section from L57 to L68? The rationale behind your study isn't entirely clear in this part. When you mention that the larval stage can retain remnants of the host plant material, are you suggesting that you consider the larval stage responsible for pollination? Additionally, how do you differentiate between pollen transfer through consumption during the larval stage and pollen dispersion through pollination during the adult phase? It would be valuable to delve deeper into the discussion on this topic as well.
Author Response
Reviewer#3:
This paper presents a highly original and captivating study on the cross-regional pollination behavior of Noctuidae moths between China and the Indo-China Peninsula. I have a few minor suggestions for improvement.
Response: We thank you for your positive feedback. We have taken all your suggestions into account and have refined the manuscript accordingly.
Q1: I recommend implementing references at L57, including other examples where identifying pollen on the body surface of insects has been demonstrated as an effective method for determining the host plant range. These examples can encompass a range of approaches, from molecular to morphological technique. I take two examples here.
Some potential examples here:
Galimberti, Andrea, Fabrizio De Mattia, Ilaria Bruni, Daniela Scaccabarozzi, Anna Sandionigi, Michela Barbuto, Maurizio Casiraghi, and Massimo Labra. "A DNA barcoding approach to characterize pollen collected by honeybees." PLoS One 9, no. 10 (2014): e109363.
Scaccabarozzi, Daniela, Kingsley W. Dixon, Sean Tomlinson, Lynne Milne, Björn Bohman, Ryan D. Phillips, and Salvatore Cozzolino. "Pronounced differences in visitation by potential pollinators to co-occurring species of Fabaceae in the Southwest Australian biodiversity hotspot." Botanical Journal of the Linnean Society 194, no. 3 (2020): 308-325.
Response: We have incorporated the recommended references into our manuscript. These additional sources enrich the context of our study and provide robust support for the methodologies we employed.
Q2: Could you also provide further clarification for the section from L57 to L68? The rationale behind your study isn't entirely clear in this part. When you mention that the larval stage can retain remnants of the host plant material, are you suggesting that you consider the larval stage responsible for pollination? Additionally, how do you differentiate between pollen transfer through consumption during the larval stage and pollen dispersion through pollination during the adult phase? It would be valuable to delve deeper into the discussion on this topic as well.
Response: We acknowledge the need for further clarification and have revised the manuscript to elaborate on the rationale behind our study methods. We explain the differentiation between larval consumption of plant material and adult pollen dispersal, which reinforces the focus of our study on the pollination potential of adult moths. We have ensured that the discussion is clear and logically presented to enhance the reader's comprehension of our research objectives and findings.